# Development of a Computational Model of the Mechanical Behavior of the L4–L5 Lumbar Spine: Application to Disc Degeneration

**DOI:** 10.3390/ma15196684

**Published:** 2022-09-26

**Authors:** Galina Eremina, Alexey Smolin, Jing Xie, Vladimir Syrkashev

**Affiliations:** 1Institute of Strength Physics and Materials Science, Siberian Branch of the Russian Academy of Sciences, Pr. Akademicheskii, 2/4, 634055 Tomsk, Russia; 2State Key Laboratory of Explosion Science and Technology, Beijing Institute of Technology, Beijing 100081, China; 3Department of General Medicine, Siberian State Medical University, Moskovsky Trakt, 2, 634050 Tomsk, Russia

**Keywords:** lumbar spine, intervertebral disc degeneration, poroelasticity, computer simulation, method of movable cellular automata

## Abstract

Degenerative changes in the lumbar spine significantly reduce the quality of life of people. In order to fully understand the biomechanics of the affected spine, it is crucial to consider the biomechanical alterations caused by degeneration of the intervertebral disc (IVD). Therefore, this study is aimed at the development of a discrete element model of the mechanical behavior of the L4–L5 spinal motion segment, which covers all the degeneration grades from healthy IVD to its severe degeneration, and numerical study of the influence of the IVD degeneration on stress state and biomechanics of the spine. In order to analyze the effects of IVD degeneration on spine biomechanics, we simulated physiological loading conditions using compressive forces. The results of modeling showed that at the initial stages of degenerative changes, an increase in the amplitude and area of maximum compressive stresses in the disc is observed. At the late stages of disc degradation, a decrease in the value of intradiscal pressure and a shift in the maximum compressive stresses in the dorsal direction is observed. Such an influence of the degradation of the geometric and mechanical parameters of the tissues of the disc leads to the effect of bulging, which in turn leads to the formation of an intervertebral hernia.

## 1. Introduction

Degenerative diseases of the lumbar spine are widespread among populations of different ages, and the problem of their treatment still remains unresolved. The intervertebral disc undergoes the greatest degenerative changes that require its replacement with a prosthesis. The complex structure of the intervertebral disc ensures the redistribution of stresses in the vertebrae [1,2]; therefore, degenerative changes in the lumbar spine affect the biomechanics of the spine as a whole. Therewith the lumbar spine (L3–S1) is most affected [3].

One of the causes of intervertebral disc degeneration (IDD) is an alteration in the supply of nutrient fluid to the intervertebral disc (IVD). The disc is nourished via the blood vessels leading through the cartilaginous endplates (CEP) by 65% along the surface of the annulus fibrosus (AF) by 35% [4,5]. At the cellular level, the loss of nutrient fluid inflow and metabolic waste outflow can create a low glucose and acidic environment for nucleus pulposus (NP) cells, thereby reducing cell viability and extracellular matrix (ECM) production. Oxygen, glucose, and lactic acid pass through the disc ECM by diffusion, supply nutrients, and waste is transported to the disc. The decrease in diffusion affects the transport of nutrients, as well as the removal of waste products from the disc. Low oxygen concentration and low pH (due to lactic acid concentration) affect disc cell metabolism, interfering with disc cell viability and limiting the ability of disc cells to synthesize new proteoglycan [6]. It is assumed that changes in the nature of nutrition are influenced by age, smoking, and genetics. With age-related changes, a change occurs in the CEP and the material of the annulus fibrosus. Cartilaginous endplates undergo sclerotic structural changes. Senescent cells in the disc gradually reduce their ability to proliferate so that their density decreases, especially in the annulus fibrosus [7]. The impact of nicotine significantly reduces the viability of biological cells in the nucleus pulposus [8]. A decrease in the synthesis of aggrecan in the NP leads to its dehydration and subsequent disruption of its mechanical function [9,10]. The dehydrated nucleus pulposus is not able to evenly distribute the mechanical pressure on all IVD structures. For this reason, the AF experiences an increased load, which can affect its structure in the form of local damage. Damage to the annulus fibrosus leads to the formation of IVD protrusions and hernias, which is the cause of acute radicular pain syndrome. Injuries significantly affect the change in the geometry of the intervertebral disc, thereby probably disrupting blood supply.

In order to study the influence of various factors on the mechanical behavior of the spine, experimental studies based on in vitro and in vivo methods are used [11]. Ex vivo studies are mainly focused on histological data and the results of magnetic resonance imaging and radiography, where it is possible to detect degeneration in the structure of the vertebrae, intervertebral discs [12,13], geometrical changes in the spine elements, as well as evaluate the kinematics of the system [14]. Testing machines are used to study the biomechanical properties of the spine [15], such as its stiffness [16] and the critical force corresponding to the onset of fracture in the system [17]. The above methods of experimental studies provide important information about the integral parameters of the spine, but they do not have the ability to determine the distribution of strains/stresses within the system under shock loads. Moreover, empirical research on some issues requires a lot of time and money and is also limited by ethical standards. In addition, degenerative disc disease is associated with progressive changes in the lumbar intervertebral discs, affecting both the mechanical characteristics of tissues and geometric parameters. In degenerated discs, various morphological changes may be present individually or, more often, in various combinations.

Computer simulation, in contrast to in vitro field experiments, makes it possible to evaluate the impact of each aspect of disc degradation individually due to the absence of limitations associated with the availability of laboratory disc samples with the necessary characteristics [18]. Most of the simulation has been performed on the basis of the finite element method using the commercial software ANSYS and ABAQUS [19]. For modeling, the mechanical behavior of the spine, mainly elastic models, is used [20,21,22,23,24], where the description of the mechanical behavior of the vertebrae and discs is reduced to the linear elastic behavior of a homogeneous material with averaged properties. However, it is known that the cancellous, CEP, AF, and NP tissue, as well as the tissues of the disc, are 70–90% composed of fluid. The redistribution of which during dynamic loading significantly affects the mechanical response of the material as a whole and is the main factor in the nonlinear behavior of biological tissues. Thus, the contribution of the fluid to the nonlinear behavior of the disc tissues may be taken into account in viscoelastic models [25,26,27,28]. Two-phase poroelastic models received the greatest distribution in the description of the mechanical behavior of the spinal column, where the influence of the fluid is taken into account implicitly [29,30,31,32]. Recently, works have begun to appear on the use of three-phase models based on the use of viscoelastic parameters of disc materials in a poroelastic medium [33,34,35] to describe the mechanical behavior of disc tissues. It is worth noting that methods based on the continuum representation of the material provide a good understanding of the kinematics of the process and the pattern of the stress and strain fields. Most of these works are devoted to the study of the possibilities of regeneration of the tissues of the disc and its surrounding tissues under the influence of physiological and other types of loads, as well as the influence of a degraded disc on the surrounding elements of the spine. However, there are few works devoted to the numerical study of the mechanisms of the formation of intervertebral hernias [6]. Most of the work on the study of disc degradation is devoted to the study of the effect of degenerative changes in cartilage [36].

In the numerical study of the mechanical behavior of the spine, as well as other complex systems, it is necessary to carry out the required work on the verification and validation of the developed model [37]. One of the important aspects of modeling in biomechanics is the determination of the strain rate sensitivity of the developed model.

The aim of this work is to develop a numerical model to simulate the mechanical behavior of the lumbar spine as a whole, including with a degenerated intervertebral disc, under physiological loading. As a modeling method, the method of movable cellular automata (MCA) was adopted, which is representative of computational particle mechanics and makes it possible to correctly describe the mechanical behavior of poroelastic heterogeneous materials, including explicit accounting for the generation and development of damage [38,39]. First of all, the developed 3D numerical model of the lumbar spine segment was verified and validated. After that, numerical studies of the lumbar spine with various degenerative changes in intervertebral discs under physiological loads were carried out. Finally, we discussed the results obtained here and compared them with available data from the literature.

## 2. Materials and Methods

### 2.1. Method of Movable Cellular Automata

In the last decades, movable cellular automata have been successfully used for modeling heterogeneous materials, including bone tissues at the macroscale and mesoscale [38,39,40,41]. In this method, the material is considered an ensemble of discrete elements of the same finite size (movable cellular automata) that interact with each other according to certain rules, which due to many-body interaction forces, describe the deformation behavior of the material as an isotropic elastoplastic body. The motion of the movable cellular automata is governed by the Newton–Euler equations for their translation and rotation:(1)$$\left\{\begin{array}{l} m_{i}{\ddot{R}}_{i}=\sum_{j=1}^{N_{i}}F_{ij}^{\mathrm{pair}}+F_{i}^{\Omega },\\ {\overset{⌢}{J}}_{i}{\dot{\omega }}_{i}=\sum_{j=1}^{N_{i}}M_{ij}, \end{array}\right.$$
where **R***_i_*, $\omega_{i}$, *m_i_*, and ${\overset{⌢}{J}}_{i}$ are the location vector, rotation velocity, mass, and moment of inertia of *i*-th element; $F_{ij}^{\mathrm{pair}}$ is the pair force of interaction of the *i*-th and *j*-th elements; and $F_{i}^{\Omega }$ is the force acting on *i*-th element due to interaction of all its neighbors with the other elements (so-called volume-dependent force). Herein, the upper dot denotes a time derivative. In the second line of Equation (1), $M_{ij}=q_{ij}(n_{ij}\times F_{ij}^{\mathrm{pair}})+K_{ij}$ is the total torque of pair *i–j*, where *q_ij_* is the distance between the center of *i*-th element and its contact point with *j*-th neighbor, $n_{ij}=(R_{j}-R_{i})/r_{ij}$ is the unit vector directed from the center of *i*-th element to the *j*-th one and *r_ij_* is the distance between centers of these elements, and $K_{ij}$ is the torque due to only relative rotation of the pair elements.

If the material is isotropic, the volume-dependent force can be written as follows:(2)$$F_{i}^{\Omega }=-A\sum_{j=1}^{N_{i}}P_{j}S_{ij}n_{ij}$$
where *P_j_* is the hydrostatic pressure in the bulk of element *j*, *S_ij_* is the interaction area of elements *i* and *j*, and *A* is the material parameter, which is determined by the ratio of the elastic moduli. By taking this into account, it is possible to rewrite the total force acting on automaton *i* as a decomposition of normal and tangential components ($F_{ij}^{n}$ and $F_{ij}^{\tau }$):(3)$$F_{i}=\sum_{j=1}^{N_{i}}\left(F_{ij}^{\mathrm{pair}}-AP_{i}S_{ij}n_{ij}\right)=\sum_{j=1}^{N_{i}}\left[\left(F_{ij}^{\mathrm{pair},n}\left(h_{ij}\right)-AP_{j}S_{ij}\right)n_{ij}+F_{ij}^{\mathrm{pair},\tau }\left(l_{ij}^{\mathrm{shear}}\right)t_{ij}\right]=\sum_{j=1}^{N_{i}}\left(F_{ij}^{n}+F_{ij}^{\tau }\right)$$
where $F_{ij}^{\mathrm{pair},n}$ and $F_{ij}^{\mathrm{pair},\tau }$ are the corresponding components of the pair interaction force depending on the central *r_ij_* and tangential $l_{ij}^{\mathrm{shear}}$ relative displacements, respectively. By using the homogenization procedure described in [39], it is possible to determine the average stress tensor in the bulk of element *i* as follows:(4)$${\bar{\sigma }}_{\alpha \beta }^{i}=\frac{1}{V_{i}}\sum_{j=1}^{N_{i}}q_{ij}n_{ij,\alpha }F_{ij,\beta }$$
where α and β are used to denote the axes *X*, *Y*, and *Z* of the global coordinate system; *V_i_* is the element volume; $n_{ij,\alpha }$ is the α component of the vector $n_{ij}$; $F_{ij,\beta }$ is the β-component of the force acting between elements *i* and *j*. Having components of the stress tensor allows easy computing of all its invariants, including von Mises stress.

Elements of the pair may represent the parts of different solids (in this case, it is a contracted pair) or of the same solid (i.e., a bonded pair, and interaction of these elements is not a real contact). The size of an element is characterized by one parameter *d_i_*, which is an approximation because geometrically, the element is determined by its areas of interaction with neighbors.

In order to characterize the deformation of element *i* due to its normal interaction with element *j,* we used the following formula for normal strain:(5)$$\xi_{ij}=\frac{q_{ij}-d_{i}/2}{d_{i}/2}$$

Each element of the pair may be made of different materials. Hence the increment of the relative displacements *r_ij_* of the pair leads to different strain increments from *i*-th and *j*-th elements:(6)$$\Delta r_{ij}=\Delta q_{ij}+\Delta q_{ji}=\Delta \xi_{ij}d_{i}/2+\Delta \xi_{ji}d_{j}/2$$
where symbol Δ denotes an increment per time step Δ*t*; the same is valid for the tangential displacement $l_{ij}^{\mathrm{shear}}$ and shear strain $\gamma_{ij}$. In order to define the strain distribution in the pair, we needed the rule for computing interaction force (constitutive equation). In MCA, it is the same as Hooke’s law for corresponding components of stress tensor:(7)ΔFijpair,n=2G(Δξij)−1−2GKΔPiΔFijpair,τij=2GiΔγij
where *K* and *G* are the bulk and shear moduli of the material of *i*-th element, *P_i_* is computed via the stress tensor components defined by Equation (4) at the previous time step and may be corrected by additional iteration.

Due to the necessity of Newton’s third law, the increments of the reaction forces of the elements *i* and *j* are calculated based on the solving of the following system of equations:(8)$$\left\{\begin{array}{l} \Delta F_{ij}^{\mathrm{pair},n}=\Delta F_{ji}^{\mathrm{pair},n}\\ \Delta \xi_{ij}d_{i}/2+\Delta \xi_{ji}d_{j}/2=\Delta r_{ij}\\ \Delta F_{ij}^{\mathrm{pair},\tau }=\Delta F_{ji}^{\mathrm{pair},\tau }\\ \Delta \gamma_{ij}d_{i}/2+\Delta \gamma_{ji}d_{j}/2=\Delta l_{ij}^{sh} \end{array}\right.$$
where Δ*r_ij_* is the change in the distance between the centers of the elements *i* and *j* per time step Δ*t*, and $\Delta l_{ij}^{sh}$ is the value of the relative shear displacement of these interacting elements. This means that discrete elements made of different materials deform differently according to their elastic properties and stress state. This ensures correct modeling of the contact interaction of different solids as well as the interaction at the interface between the inclusion and the matrix in heterogeneous materials (composites).

In order to integrate Equation (1), an explicit velocity Verlet scheme was used. In order to make the scheme stable, the value of the time step was limited by the time of sound propagation along the element.

In order to describe the mechanical behavior of the fluid-saturated material in the MCA method, the following effective (implicit) characteristics of the automata are introduced: the volume fraction of interstitial fluid (its density *ρ*), porosity *ϕ*, permeability *k*, and the ratio *a* = 1 − *K*/*K*_s_ of the macroscopic value of bulk modulus *K* to the bulk modulus of the skeleton (the solid part of the material) *K*_s_ [42]. The mechanical influence of the interstitial fluid on the stresses and strains in the solid skeleton of an element is described on the basis of the linear Biot’s model of poroelasticity [43], which assumes that the mechanical response of a “dry” element is linearly elastic, and the mechanical effect of the pore fluid on the element stress can be described in terms of the local pore pressure *P*^pore^ (fluid pore pressure in the volume of the element), which affects only the diagonal components of the stress tensor. It means that only the relations for the central interaction in Equation (7) should be modified:(9)ΔFijpair,n=2GiΔεij−aiΔPiporeKi−1−2GiKiΔPi

The fluid pore pressure in the element is calculated based on the relationships of Biot’s poroelasticity model with the use of the current density of pore fluid [43]. Linearly compressible fluid is described by the following equation of state
(10)$$\rho \left(P^{\mathrm{pore}}\right)=\rho_{0}\left(1+\left(P^{\mathrm{pore}}-P_{0}\right)/K_{\mathrm{fl}}\right)$$
where *ρ* and *P*^pore^ are the current pore fluid density and pressure, *ρ*_0_ and *P*_0_ are the equilibrium values of these parameters under standard conditions, and *K*_fl_ is the fluid bulk modulus.

The total pore space of all elements is assumed to be interconnected, which provides the possibility of redistribution (filtration) of interstitial fluid between the interacting elements. The “driving force” of filtration is a pore pressure gradient. By neglecting gravitational effects, the equation of interstitial fluid filtration in the pore space can be written as follows:(11)$$\phi \frac{\partial \rho }{\partial t}=K_{\mathrm{fl}}\nabla \left[\frac{k}{\eta }\nabla \rho \right]$$
where *η* is the fluid viscosity and *k* is the permeability of the solid skeleton, which can be calculated from the current porosity *ϕ* as:(12)$$k=\phi d_{\mathrm{ch}}^{2}$$
where *d*_ch_ is the diameter of the filtration channel.

Equations (9)–(11) were solved numerically using the first-order Euler scheme for integration in time on a mesh made of the centers of the interacting elements (by analogy to the finite volume method on an ensemble of discrete elements). According to the used approximations, there is no fluid transfer between elements if *ρ* ≤ *ρ*_0_.

The method presented above was implemented in the in-house code MCA3D, which is written in C++ programming language and utilizes the Qt library for creating models, visualization, and analyzing the simulation results. The MCA3D code has been used in many studies by the authors and their colleagues, the results of which have been published in many papers. In particular, the verification and validation of poroelastic models of the tissues of the lumbar spine based on the MCA method were carried out in [44,45].

### 2.2. Model of the Lumbar Spine Segment

The lumbar spine consists of the vertebrae (blue parts in Figure 1a) and the intervertebral discs (red part in Figure 1a). In its turn, the vertebral body consists of a cortical shell (blue in Figure 1b) and an inner part made of cancellous tissue (cyan in Figure 1b). The intervertebral disc is located between the vertebrae and has a three-component structure: the nucleus pulposus (NP), the annulus fibrosus (AF), and the cartilaginous endplates (CEP) (Figure 1b). The geometric model of the spine used herein was taken from the Internet. The creation of the cortical membrane, nucleus pulposus, cartilaginous plates, and annulus fibrosus was carried out using the open-source Free CAD software version 0.18.

The mechanical loading was applied by setting the same velocity in the vertical direction to the upper layer of the automata of the sample (arrows in Figure 1a), while the automata of the lower layer of the sample were fixed (triangles in Figure 1a). At the initial stage of loading, the velocity of the upper layer automata increased gradually from 0 to 0.1 m/s and then remained constant. This ensures a quasi-steady deformation regime, despite the dynamic governing equations. We stopped the calculations when the resistance force of the spinal segment model reached a value corresponding to the physiological load.

The poroelastic properties of the tissues of the lumbar spine used in this study are presented in Table 1 and correspond to the data from the literature [46]. The elastic properties of the tissues of the spine correspond to the data obtained experimentally in solid phase nanoindentation [47,48] for the matrix, as well as the results of tensile/compression experiments [49,50,51]. Moreover, the input parameters for our poroelastic model are in the range of parameter values most commonly used in other numerical studies [28,37,52,53,54,55,56]. The fluid in bone tissues is assumed to be equivalent to salt water with a bulk modulus of *K*_f_ = 2.4 GPa, a density of *ρ*_f_ = 1000 kg/m^3^, and a viscosity of *η*_f_ = 1 mPa s [57]. The presented model does not include any dependence of the material parameters on temperature, and the values of the used parameters correspond to room temperature.

### 2.3. Model Verification

Verification of a model is aimed at assessing the correctness and efficiency of the numerical scheme for solving the governing equations of the method. The key component of verification of a numerical model is the analysis of the convergence of the modeling results with increasing the resolution of the discrete model (in the case of particle methods, this corresponds to decreasing the particle size). Usually, the discrete representation of the model is considered optimal when a further increase in its resolution provides no more than a 5% difference in the results [53].

In this study, the convergence analysis of a three-dimensional model of the L4–L5 lumbar spine was carried out by studying the stiffness of the model and the fields of hydrostatic pressure distribution at different discretization of the considered model under compression in the vertical direction (Figure 2). Herein, the size of movable cellular automata (discrete elements) in the model varied from 0.25 to 2.0 mm. The compression of the model was carried out by setting a constant velocity of 0.1 m/s to the upper layer of the particles while the lower layer was fixed (Figure 1a).

The results on the convergence of the model stiffness showed that the dependence of the stiffness on the automata size is nonlinear, and the total scatter between the values for the minimum and maximum sizes did not exceed 10% (Figure 3). A small difference in the results between the values for automata sizes smaller than 1.5 mm indicates a good accuracy of the developed model for determining such important integral parameters as stiffness. Therefore, further numerical studies were performed on the models with an automata size of 0.5 mm.

### 2.4. Model Validation

In this work, the stiffness parameter of the lumbar spine model was validated in accordance with the available experimental data [58]. The variable parameter was the loading rate. Within the framework of the model under consideration, loading was specified along the axis of the spine in the vertical direction at a speed of 0.1 m/s, 0.2 m/s, and 1.0 m/s (corresponding strain rates are 5.5 s^−1^, 12 s^−1^, and 70 s^−1^). The obtained loading curves are shown in Figure 4a. At the loading speed of 0.1 m/s, the value of the effective stiffness of the segment of the lumbar spine reaches 1835.1 ± 645.6 N/mm. At a loading speed of 0.2 m/s, the value of effective stiffness increases to 2489.5 ± 474.1 N/mm. Under highly dynamic loads (corresponding to jumps or falls) at a loading speed of 1.0 m/s, the effective stiffness of the system increases to 6551.1 ± 2017.0 N/mm (Figure 4b).

The next step in the validation of the model was to compare the distribution of the velocity vector field with experimental and other numerical data. Loading was carried out at a force of 500 N. It is very pleasant to note that the calculated fields are in excellent agreement (Figure 5) with the stereo-radiographic experimental fields [59] and displacement fields calculated using a numerical model in the transverse plane for various physiological positions [27].

Validation of the model in terms of such local parameters as the “intradiscal pressure” [60] was carried out by comparing the obtained results with the data given in the literature for uniaxial compression simulating a standing position. According to [61], the maximum load in a vertical standing position at normal weight reaches 350–1000 N. In papers [62,63], for uniaxial compression by the force of 500 N, the calculated intradiscal pressure is reported as about 0.28–0.74 MPa. According to the authors of papers [64,65,66], it ranges from 0.42 up to 1 MPa at a normal weight. For our model loaded by the force of 500 N, the pressure value in IVD is in the range of 0.2–0.72 MPa (Figure 6a). According to our results, the intradiscal pressure reaches up to 0.8 MPa under uniaxial compression with a force of 1000 N (Figure 6b). Thus, the presented model provides the intradiscal pressure values corresponding to the literature data in a wide range of loads in a vertical standing position.

## 3. Results

Simulation of Physiological Loading of Degenerated Segment of Lumbar Spine

Depending on the degree of degradation, the disc height, permeability, porosity, and elastic modulus change [67,68]. In its turn, the stress state of the system changes significantly.

In this work, at the first step, the influence of changes in the height of the intervertebral disc was studied in accordance with the data presented in [69]: in a healthy segment of the spine, the height of the intervertebral disc is about 10 mm (Figure 7a); with degradation of the disc of the first degree, it becomes 8 mm (Figure 7b); with degradation of the disc of the second degree, it is 5 mm (Figure 7c). The loading force was 500 N.

In the second step, the effect of degradation of the poroelastic properties of the disc on the stress state was studied. In accordance with the data of [26] to describe the material model at different stages of degradation of AF and NP, the parameters given in Table 2 were used.

In the third step, the influence of both changes in geometric characteristics and poroelastic properties of intervertebral disc tissues was studied.

The obtained results indicate that at the initial stages of degradation of the intervertebral disc, which is expressed in a decrease in the height of the disc by 25%, an increase in the value of intradiscal pressure occurs, combined with a slight shift of the area of maximum stresses in the dorsal direction (Figure 8, in comparison with Figure 6a). When the disk height decreases by 50%, the maximum compressive stresses shift significantly from the NP to the dorsal direction (Figure 8b), which, in turn, leads to a redistribution of the load that is different from a healthy joint. It was approved in work [70] that the NP ceases to function as a compressive stress concentrator, and compressive stresses occur in the AF and CEP. Due to such distribution of stresses, bulging occurs; therefore, the risk of a posterior disc protrusion and a hernia formation increases.

In the case of degenerative changes taken into account by changing the poroelastic properties of the tissues at the initial stage of disc degradation (AF1 and NP1 in Table 2), the maximum values of compressive stresses increase, and the area of maximum stresses shifts in the dorsal direction (Figure 9, in comparison with Figure 7a). When the properties correspond to the last stage of degradation (AF2, NP2), a shift of the local maximum of compressive stresses in the dorsal direction is also observed (Figure 9b), similar to that in the case of a change in the height of the intervertebral disc (Figure 9b) but with lower maximum stress.

With a combination of degenerative changes in the geometric parameters of the disc and the poroelastic characteristics of the disc tissues, both an increase in the amplitude of compressive stresses (Figure 10a) and the effect of displacement of the maximum stress in the dorsal direction (Figure 10b) are observed similar to those in the case of a change only in the height of the intervertebral disc.

In order to analyze in detail the effect of the disc degradation on the biomechanics of the disc itself under loading, Figure 11 depicts only the disc configuration (other parts of the spinal segment are removed) and the field of the pressure of interstitial fluid in its pore space.

One can see that the maximum fluid pressure in the pores of a healthy intervertebral disc is observed in the area of cartilaginous plates (Figure 11a); thus, the nucleus pulposus is evenly nourished through the vessels of the cartilaginous plates, similar to shown in [36]. With the degradation of the poroelastic properties of the AF and the NP, the maximum pore fluid pressure is observed in the nucleus pulposus (Figure 11b), thereby disrupting the flow of nutrients into the intervertebral disc. With degradation expressed through a change in heights of the intervertebral disc, the maximum values of the pore fluid pressure were observed in the material of the nucleus pulposus (Figure 11d). Similar changes were observed (Figure 11f) when two factors of the disc degradation were taken into account simultaneously (changes in the poroelastic parameters and the changes in the disc height). With further degradation of the disc, a shift of the area of the maximum values of the pore fluid pressure in the dorsal direction was observed, thereby contributing to disc buckling (Figure 11c,e,g).

## 4. Discussion

In the present paper, the mechanical implications of specific degenerative characteristics present in different combinations were evaluated by means of modeling using the movable cellular automaton method. The simulation results showed that when degenerative changes in the disc are taken into account only by the poroelastic properties of the materials of the nucleus pulposus and annulus fibrosus, at the first stage of degradation, an increase in the maximum intradiscal pressure with a slight shift of its area in the dorsal direction was observed; at the second stage, a pronounced shift of the maximum compressive stresses in the dorsal direction takes place with a decrease in the values of maximum stresses. When taking into account degenerative changes in the intervertebral disc through a change in its height, it was found that a decrease in the height by 25% leads to an increase in the area of the maximum stresses; in addition, there is a slight shift of this area in the dorsal direction. When the height changes by 50%, a significant shift of the area of the maximum stresses in the dorsal direction is observed. With a combination of changes in the geometric and mechanical characteristics of the intervertebral disc, there is a tendency for the area of maximum compressive stresses to shift in the dorsal direction with an increase in their values.

Most modeling works note that when studying degenerative changes in intervertebral discs, an increase in the amplitude of intradiscal pressure is observed. However, no information is provided on the area of maximum compressive stresses. Paper [71] reports a nonlinear (sinusoidal) dependence of the maximum compressive stresses in the nucleus pulposus on the degree of degradation. Therefore, at mild degrees, there is an increase in the amplitude of compressive stresses; at a second degree, a decrease in amplitude; at a third degree, its increase is observed. Similar results were obtained in [68], where it was found that at the first stages, there is an increase in hydrostatic and fluid pressure in the pores of the pulpous nucleus; at the subsequent stages of disc degradation, expressed by a change in geometric parameters or the material properties, there is a decrease in the amplitude of compressive stresses in the nucleus pulposus with an increase in compressive stresses in the annulus fibrous rings. The article [72] draws similar conclusions. In a healthy disc, the maximum compressive stresses are observed in the nucleus pulposus, and a minimum is observed in the material of the annulus fibrosus. At the first stages of disc degradation, there is an increase in compressive stresses in the nucleus pulposus, as well as an expansion of the zone of maximum compressive stresses. At the last stages of disc degradation, a significant decrease in the amplitude of compressive stresses in the disc is observed with the same amplitude in the material of the annulus fibrosus. Thus, it was shown that in the last stages, the maximum compressive stresses shift into fibrous tissues.

However, in the works described above, there is no information about the direction in which the maximum compressive stresses are shifted. In addition, a significant simplification of the geometry of model samples in finite element models leads to non-physiological stress distribution. Therefore, defining a more realistic internal and external geometry of the IVD can significantly improve the quality of the simulation results of the corresponding FE model, and it is likely that a more realistic geometry leads to more accurate stress distribution in the IVD [73].

In work [36], it was found that fluid pore pressure in NP increase with increasing CEP sclerotic conditions. At equilibrium, fluid pressure in NP must be close to zero, as there is no fluid exchange. In the presence of a less permeable path through the CEP and the OP, fluid is trapped and slowly percolates transversely through the AF [74]. Increased fluid pressure in the LR can lead to bulging of the posterior outer disc with increased vertical loading [75]. In addition, CEP may experience greater transverse shear and tensile stress near the high fluid pressure in NP. Stress in NP and CEP increases in both cases of sclerotic CEP [76]. The fluid pressure in the OP increases as the CEP calcifies, while its permeability and porosity decrease. These calculated predictions support the hypothesis that sclerotic CEP would lead to a decrease in the nutrient fluid in the NP and protrusion of the NP through the AF under compressive and rotational loads. The lack of nutrient fluid and the accumulation of waste metabolites can lead to excessive apoptosis of NP cells and loss of extracellular matrix [77]. Therefore, flow-restricted CEP calcification causes NP cell death, loss of NP ECM, and IVD collapse, resulting in a reduction in height and hence leading to IDD.

The works [78,79] show that during degenerative changes in the nucleus pulposus, stresses are redistributed in the material of the cartilaginous plates, and they cease to perform their functions. Previous reports indicated that early IVD degeneration is characterized by loss of water and proteoglycans in the NP, the inner region of the disc, which causes a decrease in intradiscal pressure (a decrease of more than 30%), followed by mechanical instability. A decrease in intradiscal pressure leads to a decrease in disc height and a redistribution of stress in the annulus [80].

The results of our study show that during degenerative changes in the initial stage, the pore pressure of the fluid in the cartilage plate decreases, while in NP and AF, it increases. Thus, the natural conditions for the supply of nutrients to the intervertebral disc are violated; in addition, such processes can lead to the degradation of cartilage plate tissues. Thus, it was shown that a decrease in the flow of nutrients in the disc tissue could be affected not only by the degradation of cartilage plates but also by changes in the height of the NP and AF components and their poroelastic properties. The data obtained require further research on what is the root cause of disc degradation: a change in the properties of cartilage plates or degradation of the NP and AF.

Experimental studies indicate a displacement of the material “bulging” of the nucleus pulposus in the dorsal direction, which in turn leads to protrusion and the formation of a hernia [81].

Our studies based on computer simulation show that during degenerative changes, the maximum stresses shift in the dorsal direction, which in turn is a precursor of hernia.

It is worth noting that our study has a number of limitations. Firstly, only the L4–L5 segment of the spine is considered. This approach does not give an idea of the impact of intervertebral disc degradation on neighboring segments. However, modeling the entire spine or lumbar region is very difficult due to the limited computational capabilities. Secondly, only the disc degradation is considered, and it is expressed only as a change in the poroelastic properties of the annulus fibrosus and the nucleus pulposus, as well as a change in the geometric thickness of the disc. The mechanical and geometric parameters of the cartilaginous plates remain the same. In addition, when creating a model of a degraded disc, the presence of osteophytes is not taken into account. Avoiding these limitations is planned to be performed at the next stage of the study.

## 5. Conclusions

The presented numerical model was used for studying the mechanical behavior of the L4–L5 lumbar spine segment under physiological loading (standing). For the analysis, the method of movable cellular automata was used. The models of the elements of the lumbar spine were validated using uniaxial compression by applying loads of 500 and 1000 N. For this, the results obtained from numerical calculations were compared with experimental data from the literature. The analysis of such a comparison showed a good quantitative agreement of the calculated data with the experimental results.

Analysis of the distribution of hydrostatic pressure showed that at the initial stages of degenerative changes, there is an increase in the amplitude of maximum compressive stresses, as well as an increase in the area of the maximum compressive stresses. When modeling large degenerative changes, a shift of the maximum compressive stresses in the dorsal direction is observed, which, as expected, can lead to disc protrusion and hernia formation. Analysis of the distribution of fluid pressure showed that at the degenerative changes, in the cartilaginous plates, a minimum value of the fluid pressure is observed, which means that the cartilaginous plates cease to play the role of an element conducting nutrients to the nucleus pulpous. Such an influence of deterioration of the properties of the AF, NP, and their height contributes to further degradation of the disc.

Thus, it was shown that disc degradation could be affected not only by the degradation of cartilaginous plates but also by changes in the height of the NP and AF components and their poroelastic properties.

In summary, we can conclude that the developed herein numerical model of the L4–L5 lumbar spine segment allows simulating the mechanical behavior of this biological system under physiological loading, taking into account degenerative changes in the intervertebral disc.

## Figures and Tables

**Figure 1 materials-15-06684-f001:**
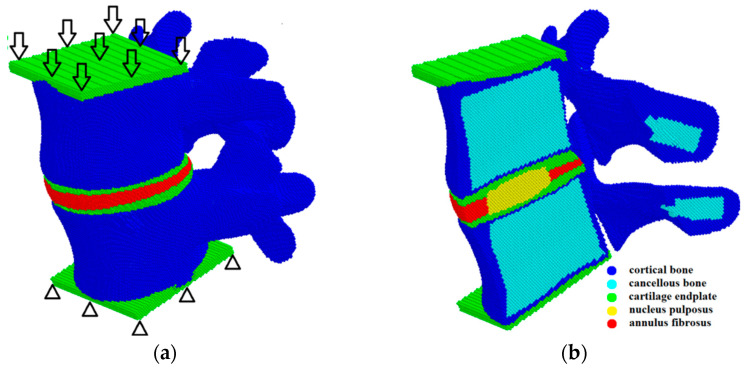
Model sample of the L4–L5 spinal segment presented as a package of automata: general 3D structure and a scheme of loading (**a**); mid-sagittal cross-section (**b**).

**Figure 2 materials-15-06684-f002:**
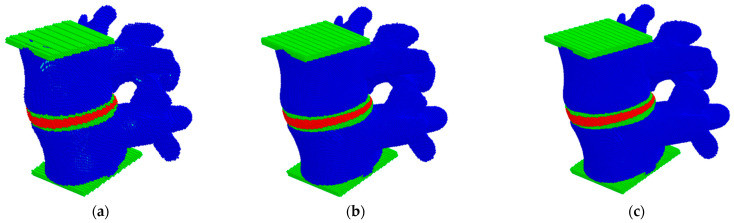
Models of the L4–L5 lumbar spine segment presented as a package of automata with different automata sizes: 2 mm (**a**), 0.5 mm (**b**), 0.25 mm (**c**).

**Figure 3 materials-15-06684-f003:**
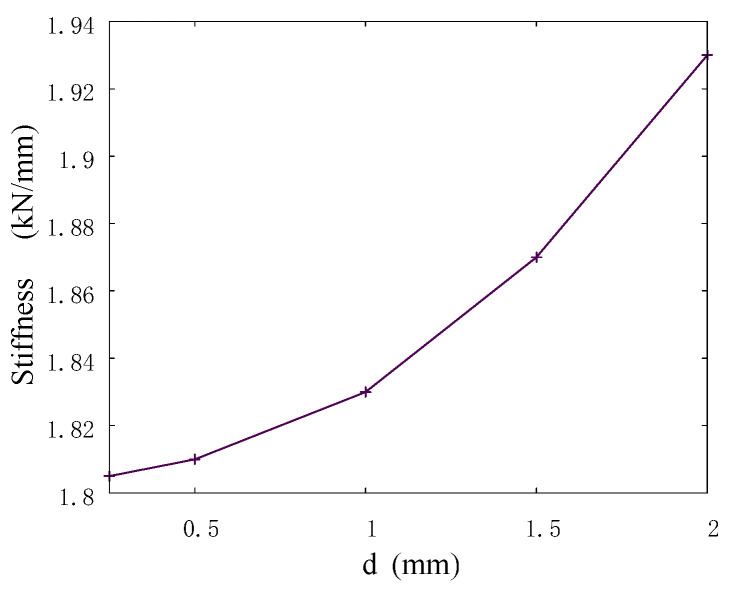
Stiffness versus size of automata in the model sample.

**Figure 4 materials-15-06684-f004:**
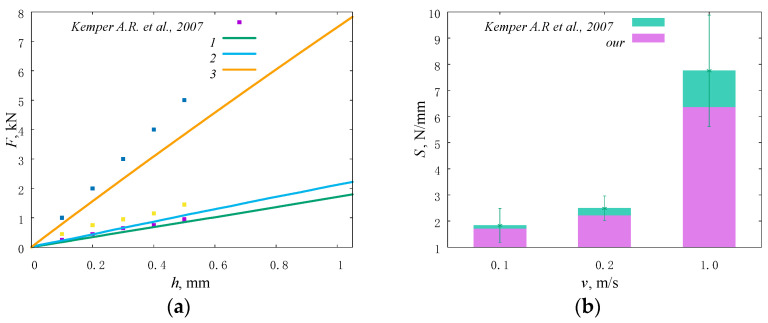
Stiffness of the model and lumbar intervertebral discs [58] under uniaxial compression at different loading rates presented as plots of force versus displacement (**a**), numbers correspond to loading speed of (1) 0.1 m/s, (2) 0.2 m/s, (3) 1.0 m/s; a histogram (**b**).

**Figure 5 materials-15-06684-f005:**
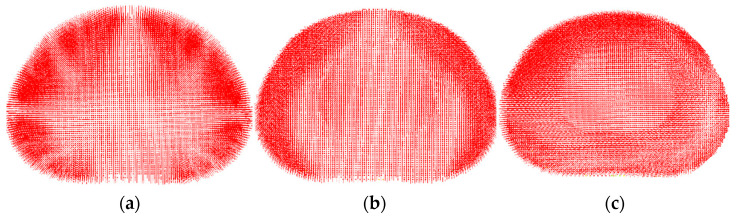
Displacement field (velocity vectors) in the central horizontal section of the intervertebral disc for the different physiological positions: standing (**a**), flexion (**b**), bending (**c**).

**Figure 6 materials-15-06684-f006:**
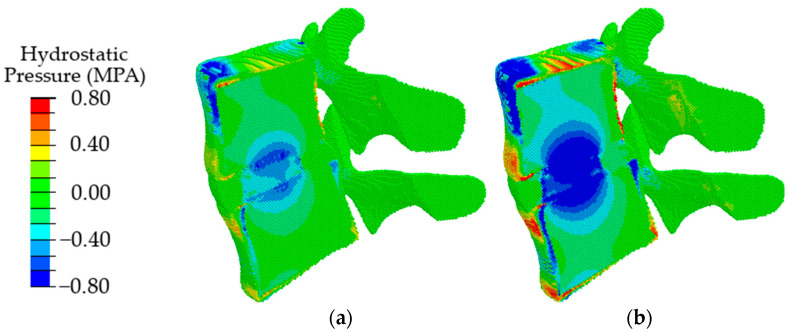
Fields of hydrostatic pressure in the spinal segment under uniaxial compression with a force of 500 N (**a**) and 1000 N (**b**).

**Figure 7 materials-15-06684-f007:**
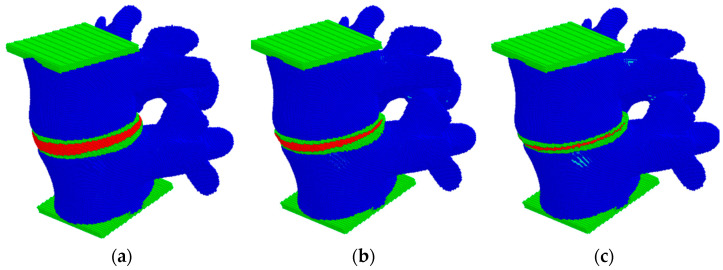
View of the model sample at different heights of the intervertebral disc: (**a**) healthy, (**b**) degenerative changes of the first degree, (**c**) degenerative changes of the second degree.

**Figure 8 materials-15-06684-f008:**
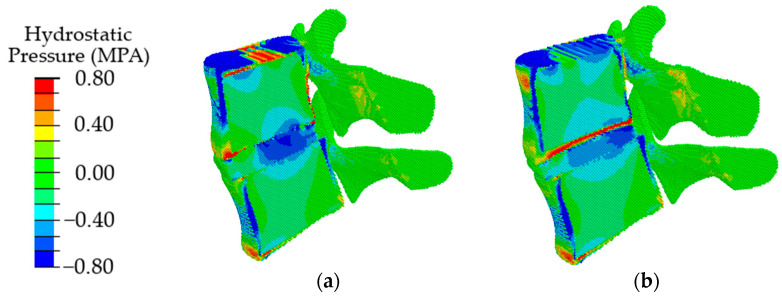
Fields of hydrostatic pressure in the spinal segment with disc degradation, which is expressed in its height at different degrees of degradation: the first degree (**a**) and the second degree (**b**).

**Figure 9 materials-15-06684-f009:**
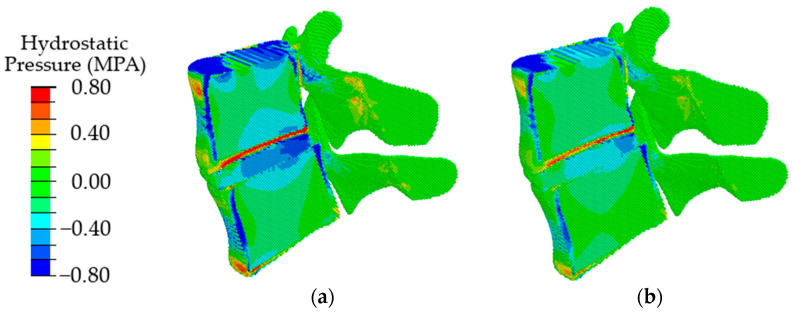
Fields of hydrostatic pressure in the spinal segment with disc degradation that are taken into account by changing the parameters of the disc materials at different stages of degradation: the first stage (**a**) and the second stage (**b**).

**Figure 10 materials-15-06684-f010:**
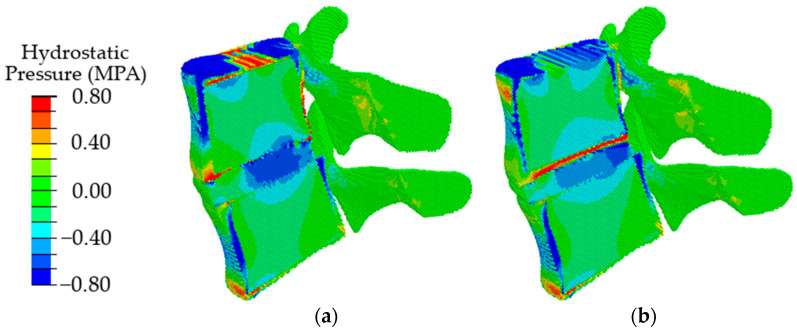
Fields of hydrostatic pressure in the spinal segment with disc degradation, which is expressed in a combination of the geometric and poroelastic characteristics of the disc at different stages of degradation: the first stage (**a**) and the second stage (**b**).

**Figure 11 materials-15-06684-f011:**
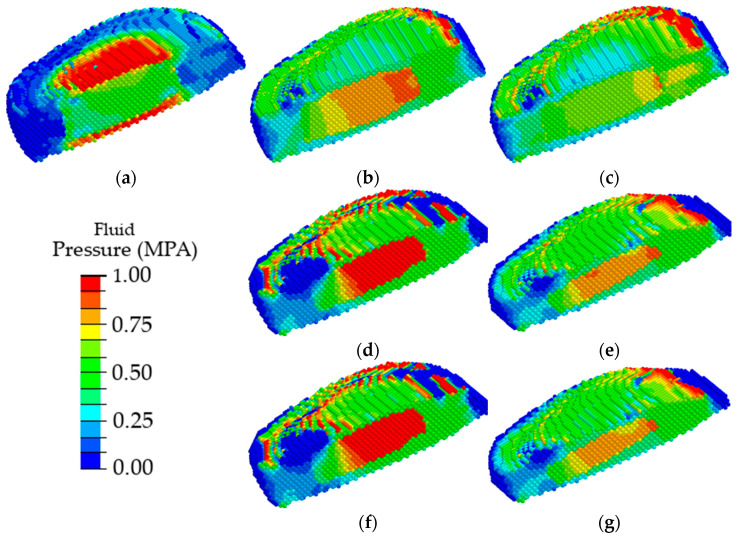
Fields of pore fluid pressure in the intervertebral disc in a healthy spinal segment (**a**) and a spinal segment with pronounced disc degradation through the parameters of the disc materials (**b**,**c**), geometric parameters (**d**,**e**), and a combination of material and geometric parameters (**f**,**g**) at different stages of degradation: the first stage (**b**,**d**,**f**) and the second stage (**c**,**e**,**g**).

**Table 1 materials-15-06684-t001:** Elastic and poroelastic parameters of the tissues.

Type of Tissue	Density of the Matrix, *ρ*, kg/m^3^	Young’s of the Matrix, E, GPa	Poisson’s Ratio ν	Bulk Modulus of the Solid, Ks, GPa	Porosity, *θ*	Permeability *k*, m^2^
AF	1060	0.0025	0.20	3.4	0.8	3 × 10^−19^
NP	1060	0.0015	0.30	5	0.8	3 × 10^−19^
CEP	1000	0.005	0.46	9	0.8	7 × 10^−18^
Cortical	1850	10	0.30	17	0.04	1 × 10^−16^
Cancellous	700	0.1	0.20	10	0.7	1 × 10^−19^

**Table 2 materials-15-06684-t002:** Elastic and poroelastic properties of degradation disc.

Stage of Degradation	Young’s Modulus *E*, MPa	Porosity *θ*	Permeability *k*, m^2^
AF 0	2.5	0.82	1.52 × 10^−19^
AF 1	2.7	0.72	1.25 × 10^−19^
AF 2	2.9	0.62	1.00 × 10^−19^
NP 0	1.5	0.78	2.42 × 10^−19^
NP 1	1.7	0.68	2.20 × 10^−19^
NP 2	2.0	0.58	2.00 × 10^−19^

## Data Availability

The data that support the findings of this study are available from the corresponding author upon reasonable request.

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
