# Peer review of "Development of a Computational Model of the Mechanical Behavior of the L4–L5 Lumbar Spine: Application to Disc Degeneration"

_materials, 2022, doi:10.3390/ma15196684_

Round 1
Reviewer 1 Report
This is very important study that enable to provide more precise effect of related factors to develop the formation of intervertebral disc hernias. However, some issues should be clarified:
1. Please provide the aim of your study in the abstract, thus, the conclusion can be related with the aim of study
2. Introduction (lines 63-75): Paragraph 3 shows the lack of previous works that have been done to solve the problem of intervertebral hernias, while paragraph 4 is the proposed novelty solution together with the aim of study. Please try to merge/rewrite those two paragraphs
3. Line 73: The first is mention, but there is no second and others
4. Minor: please manage an alignment of the note of Figure 1, and title of (b) do you mean mid sagittal cross-section?
5. Major: Result 3.3 (line 157). There is no detail explanation about stimulation of physiological loading in the method section. All result should well be described in the method section. In addition, your aim of study (in the Introduction) is to develop a numerical model to simulate the mechanical behavior of the lumbar spine as a whole, including with a degenerated intervertebral disc, under physiological loading, thus describe the procedure of those three steps of your study to show mechanical behavior stimulation:
· Please mention the procedure of first step: the influence of changes in the height of the intervertebral disc
· Please mention the procedure of the second step: the effect of degradation of the poroelastic properties of the disс on the stress state
· Please mention the procedure of the third step: the influence of a combination of changes in geometric characteristics and poroelastic properties of intervertebral disc tissues
6. Conclusion: please rewrite the conclusion and try to make a concise and straight forward conclusion related to the aim of your study in order to avoid the important message of this study. Future direction/recommendation based on this study can be describe following the conclusion.
7. Is there any limitation of study? Please describe in the end of discussion section

Reviewer 2 Report
The present manuscript is entitled ‘Development of a Computational Model of the Mechanical Behavior of the L4-L5 Lumbar Spine: Application to Disc De-generation’ and presents the development of a discrete element model for simulating the mechanical behavior of the L4-L5 under different degeneration grades. (from healthy IVD to severe degeneration) The results of modeling showed that at late stages of disc degradation, decrease of intradiscal pressure and maximum compressive stresses are observed.
Generally, the present manuscript is well written. The authors present an extensive numerical approach of a complex problem. Even though the international literature has a lot of data and results on numerical approaches of human lumbar spine, the manuscript is of interest.
The manuscript needs revisions, and the authors should address the following remarks:
1. 1. The whole numerical approach was fulfilled using the material properties of Fan R.X et al. [26] The authors should argue, search in the literature, that the utilized mechanical properties are equivalent to other studies as well. If the authors cannot achieve this comment, their study is more qualitative, and the results of the study cannot be generalized
2. 2. According to the authors the load is imposed under a constant velocity of 0,1m/s. The authors should argue if their studying the dynamic phenomenon of the problem or their discussion is limited to the static problem.
3. 3. Concerning the sensitivity analysis on the dimensions of the finite element, the authors should argue that the stiffness is also measure through an experimental study. In that case the authors will argue that the value of their prediction of stiffness is realistic, and their model can predict accurately this complex behavior
4. 4. The authors should mention and argue in their manuscript, the type of interaction between the different interfaces and materials.
5. 5. The authors should add and discuss in the manuscript horizontal section in-between the thickness of the intervertebral disc in order to see in more detail and change of stress concentration
6. 6. In section 4 the authors should add and discus the deformed shape of their models under the different degenerative changes. The reviewer can see from the attached images that a small rotation is occurring, but the authors have to argue on that.
7. 7. Finally, the authors should clarify which results were obtained from Abaqus and which results from ANSYS and add the software in the literature.
Summarizing the manuscript needs major revision before publishing.
Reviewer 3 Report
Review of “Development of a Computational Model of the Mechanical Behavior of the L4-L5 Lumbar Spine: Application to Disc Degeneration”
The authors present a computational study about the assessment of the degenerative effects on the biomechanics of a lumbar spine unit under physiological compressive conditions. The degenerative effects on the local stresses are presented.
The reviewer has several issues about the constitutive representation along with the physical reality of the described mechanisms both being the prerequisite of an accurate prediction. The following issues must be properly addressed:
1) The section “2.1. Method of movable cellular automata” is too general and must be specified to the present problematic. It is not necessary for example to introduce plasticity and fracture consideration. It is strongly recommended to re-write this section and to clearly introduce the constitutive equations of the poro-elasticity and the fluid transfer.
2) The main drawback lies in the constitutive representation given to the disc tissues. The constitutive modeling was performed within pure elasticity without considering intrinsic viscosity and nonlinearity (hyperelasticity). This is clearly a weakness of the model. These key parameters were considered in some computational models of the recent literature showing the combined effects of hydration and intrinsic viscosity on the strain rate sensitivity.
3) Another drawback lies in the way that degenerative effects are considered. The authors propose to introduce 3 discrete values (the elastic modulus, the porosity and the permeability) as direct inputs into the model in the aim to describe 3 stages of degradation. Behind the fact that this approach seems too naïve, there is no mention of nutrients supply, ph… Clearly, the mechanobiological aspects are poorly described and the biological degradation is only implicitly described. It would be welcome to clearly indicate this aspect and to properly discuss it. Moreover, the degeneration-induced evolution of the model inputs must be justified.
4) Moreover, since the degradation effects are considered only for the disc tissues, the presentation of results is not accurate and optimal. It would be better to avoid to show the vertebrae and to show only the local fields inside the disc.
5) Another major drawback concerns the verification of the model predictions. Can the authors quantitatively compare the overall load-displacement curves with (in-vitro) experimental results? Or even the local displacement fields with exiting MRI data? The authors are invited to consult the recent literature on this subject.
6) A limitation section should be added. You need to acknowledge the limitations of the constitutive modeling of the different unit spine components.
7) In the introduction section, the authors state that “One of the important aspects of modeling in biomechanics is the determination of the strain rate sensitivity of the developed model.” Nonetheless, the model predictions were not presented at different strain rates.
8) The loading parameters (in terms of velocity and temperature) should be indicated.
9) The authors can take advantage of their quantitative results to provide a connection of the local fields and the different key factors governing the intervertebral disc response such as the loading parameters, the disc structure and the mechanism of fluid transfer.
10) It is strongly recommended to enlarge the background. The list of references does not reflect the current state of the art. Especially the most recent models developed by parallel research groups are missing. Some relevant modes are useful to analyze the local mechanisms such as bulging. A discussion about the different models would be better in order to better justify the choices made by the authors. In order to enlarge the background, see for example:
“Numerical implementation of an osmo-poro-visco-hyperelastic finite element solver: application to the intervertebral disc. Computer Methods in Biomechanics and Biomedical Engineering 5, 2020, 538-550.”
“A microstructure-based model for a full lamellar-interlamellar displacement and shear strain mapping inside human intervertebral disc core, Computers in Biology and Medicine, 135, 2021, 104629.”
“Experimental and computational comparison of intervertebral disc bulge for specimen-specific model evaluation based on imaging. Frontiers in Bioengineering and Biotechnology 9, 2021, 661469.”
It is expected that the authors clearly discuss the position of this work versus other ongoing researches on this field. In particular, acknowledging the differences with recent works of parallel research groups is important.
Round 2
Reviewer 1 Report
Thank you for your effort to improve this interesting study
Author Response
Thank you for understanding and high assessment of our efforts.
Reviewer 2 Report
The manuscript could be accepted
Author Response

(The authors gave the same response as above.)

Reviewer 3 Report
I thank the authors for their responses to my previous issues. A signification modifications have been done by the authors. The new version is much better interesting, especially the new discussions and the local analysis on the disc.
I recommend the paper for publication. Before the final publication, please carrefully re-read the paper. Their are still several misprints, as for example:
- Page 4, Eq. (1) is misprinted.
- Page 15: "Analysis of the distribution of fluid pressure showed that at the at the degenerative changes" should be "Analysis of the distribution of fluid pressure showed that at the degenerative changes"
